

# How long is too long? Examining waiting times and stress in human-elevator interaction

Masahiro Shiomi[1], Masayuki Kakio[2] and Takahiro Miyashita[1]

[1] Advanced Telecommunications Research Institute International, Seika, Kyoto, Japan
[2] Mitsubishi Electric Corporation, Tokyo, Japan

## ABSTRACT

Elevators have become an indispensable infrastructure in modern societies. Our use of them involves two distinct waiting periods: the *waiting time* (before the elevator arrives) and the *transit time* (riding to a destination). Reducing both is crucial for enhancing the user experience. For this purpose, researchers have focused on two approaches: such technological schemes as sophisticated elevator control algorithms for minimizing the actual waiting times, and such cognitive approaches as positioning mirrors that divert users' attention for reducing the perceived waiting times. However, past studies failed to sufficiently examine the relationship between the actual and perceived waiting times in real elevator usage conditions; nor have they investigated how these waiting times are related to user stress. In this study we addressed these relationships by conducting an experiment that replicated real-world elevator usage scenarios and gathered both self-reported perceived waiting times and stress data from participants and objectively measured the actual waiting times by video analysis. Investigation of these data clarified the discrepancy between actual and perceived waiting times and more deeply explored how waiting times influence user stress.

# INTRODUCTION

Elevators are an indispensable infrastructure for intra-facility transportation in modern societies (*Al-Kodmany, 2023a*; *Glaeser, 2020*). As buildings grow larger and taller, the functions required of elevators have become increasingly diverse. In addition to mechanical innovations that enhance the capacity for transporting people and goods (*Al-Kodmany, 2023b*; *Zrnić et al., 2023*), research has focused on efficiently coordinating multiple elevators, treating them much like multi-agent systems, and on resource management that considers user behaviors *via* both simulations and collected data in real environments (*Bernardini et al., 2021*; *Hakonen & Siikonen, 2008*; *Maleki, Bhatta & Mashayekhy, 2023*; *Vodopija et al., 2022*).

With such advances in elevator functionality and performance, understanding user experiences has become more crucial. In such typical transportation infrastructure services as buses, subways, and airplanes, waiting times have emerged as a significant factor in user

Corresponding author
Masahiro Shiomi, m-shiomi@atr.jp

evaluations (*Ayodeji, Rjoub & Özgit, 2023*; *Feng et al., 2016*; *Ibrahim et al., 2020*). These studies reported that perceived waiting time is pivotal in shaping overall satisfaction. Beyond transportation, studies on internet-based services and health/commercial services also examine waiting-time evaluations (*Bielen & Demoulin, 2007*; *Nah, 2004*), confirming that time-related user experiences are crucial indicators of service quality.

However, this topic remains underexplored for elevators, even though they are vital to every day indoor mobility. Although the above past studies proposed various methods to reduce elevator waiting times, fundamental knowledge remains limited concerning how long users actually wait and how long they perceive such waiting times. In the context of elevator use, two distinct waiting phases are crucial: *waiting time* (the time when a passenger waits in the lobby for an elevator) and *transit time* (the time spent by a passenger traveling in the elevator to her destination floor) (*Chartered Institution of Building Services Engineers (CIBSE), 2020*). Although previous work suggested that objective waiting durations often deviate considerably from perceived durations (*Fan, Guthrie & Levinson, 2016*), related past studies did not focus on empirical comprehensive investigations of both *waiting time* and *transit time* in actual elevator scenarios.

Moreover, while it is generally assumed that longer waiting induces greater stress (*Osuna, 1985*; *Rankin, Sweeny & Xu, 2019*; *Suck & Holling, 1997*), few studies have quantitatively examined this relationship in such a commonplace yet critical setting as elevator use. Although a past study investigated perceived stress on an online survey with different combinations of *waiting time* and *transit time* (*Bird et al., 2016*), it also remains unclear whether perceived stress differs between waiting for an elevator to arrive and riding it in actual settings. Clarifying both the discrepancy between perceived and actual waiting times and how waiting times impact stress can provide essential guidelines for elevator design.

Based on these considerations, this study investigates two research questions concerning perceived waiting times and stress during elevator use:

RQ1: To what extent do perceived waiting times diverge from actual waiting times in elevator use?

RQ2: How do elevator waiting times affect perceived stress, and in what ways might this relationship vary?

To address these questions, we conducted a data collection in a real elevator functioning in a commercial building. Participants were observed and surveyed while waiting for and riding the elevator, enabling us to collect actual waiting times, perceived waiting times from self-reports, and stress levels by questionnaires. We then analyzed these factors to elucidate their interrelationships. The contributions of this study are twofold: (1) we quantitatively clarify the discrepancy between perceived and actual waiting times in elevator use cases, and (2) we model how perceived stress correlates with waiting times related to elevator use, thereby facilitating the prediction of stress responses based on waiting times and types.

The remainder of this article is structured as follows: "Related Work" details related research and positions this study, "Materials and Methods" explains our data collection

process, "Results" presents the experimental results, "Discussion" discusses the implications of these findings, and finally, "Conclusions" concludes the article.

## RELATED WORK

### Time perception

According to past studies, various factors influence time perceptions, such as variety and types of different stimuli (*Buffardi, 1971*; *Mo, 1975*), subjective emotions (*Droit-Volet & Meck, 2007*; *Tipples, 2008*), stimulus types, temporal paradigms (*Cui et al., 2023*), age (*Hancock & Rausch, 2010*), and gender (*Hanson & Buckworth, 2016*). These studies reported that negative emotions increased the perceived times more than positive emotions, and word stimuli showed relatively smaller distortions of time perception compared to visual and sound stimuli. From another perspective, a past study reported that even metabolic rates delay time-flow perceptions (*Oshakbayev et al., 2024*). In addition, several studies focused on the cognitive models of time estimation (*Zakay & Block, 1996*), and reported that prospective judgments are longer and less variable than are retrospective judgments (*Block & Zakay, 1997*).

These studies provided the essential basic knowledge to understand what factors affect time perception, establishing critical guidelines for experimental design for related works that focus on time perception. On the other hand, since the main purpose of this series of studies was to verify the impact of detailed factors on time perception under strictly controlled conditions, their research aims are different from ours. We focused on directly verifying the relationship between waiting times and stress when using elevators.

### Relationships between waiting times and stresses

Various empirical studies have been conducted on how the unavoidable waiting times for various services affect satisfaction with service and stress. For example, a past study investigated the perceived waiting times reported by passengers at various types of transport stops and stations and compared it with the actual waiting times measured by recorded videos; the perceived waiting times exceeded the actual waiting times (*Fan, Guthrie & Levinson, 2016*). Another study investigated different kinds of transportation services (*Ayodeji, Rjoub & Özgit, 2023*) and reported that passenger satisfaction with waiting at airports is essential for customer loyalty in long-term sustainability. Waiting times strongly influence such different services as fast-food restaurants and health services (*Abdul Hamid, 2021*; *Bielen & Demoulin, 2007*; *Pruyn & Smidts, 1998*). From another perspective, a past study reported that for web users, the tolerable waiting times for information retrieval is about 2 s (*Nah, 2004*). Similar to such human-computer interaction, people perceived more stress when the response time of a conversational robot exceeds 2 s (*Shiwa et al., 2009*), although overly hasty response times are also perceived as strange by people in interactions with anthropomorphized robots (*Shiomi et al., 2018*; *Shiwa et al., 2009*).

Improving satisfaction with waiting environments is one effective way to decrease the perceived stress caused by waiting times (*Bielen & Demoulin, 2007*; *Nah, 2004*). One traditional and famous approach related to satisfaction while waiting for an elevator is to

install a mirror (that diverts the attention of those who are waiting) in an elevator hallway (*Sasser et al., 1978*). Another study placed television sets in waiting environments and described the customer satisfaction with their waiting environments (*Pruyn & Smidts, 1998*). Such approaches are useful and effective in real service contexts.

However, although these past studies investigated the relationships between waiting times and stress related to satisfaction in various service settings, they neglected situations where passengers wait or take elevators. Nor has the relationship between their waiting times and perceived stress in such situations received adequate focus.

### Human-elevator interaction

Some studies focused on understanding how people use elevators. For example, a past study analyzed the in-cabin behavior of passengers using multiple sensor devices and identified the preferred standing positions and movement patterns in elevators (*Robal et al., 2022*). Another study investigated preferred standing positions in elevators through a web-based survey to design acceptable behaviors of mobile robots that share elevators with humans (*Gallo et al., 2022*). From an engineering perspective, researchers developed voice-based control functions and investigated voice-based interactions with smart elevators to improve user experiences (*González-Docasal et al., 2024*; *Meenatchi, Aishwarya & Shahina, 2016*; *Shiomi, Kakio & Miyashita, 2024*).

These studies provided knowledge that contributed to achieving more natural and smooth interaction between passengers and elevators in a social context, although they focused less on understanding the gaps between perceived waiting times and actual waiting times, as well as the relationships between these times and perceived stress. Another study investigated the relationships among *waiting time*, *transit time*, and perceived stresses in elevator use cases; unfortunately, their experiment was an online survey and focused on specific lengths of waiting times (*Bird et al., 2016*). Therefore, the relationships between waiting times and perceived stresses in actual settings remain unknown.

### Summary of our study

Our current study investigated the above two research questions through an empirical study where participants rode an actual elevator in a real building. First, we analyze the gap between perceived and actual waiting times in two situations: waiting for the elevator and riding it. Then we analyzed the relationships between the perceived stress and perceived/actual waiting times.

## MATERIALS AND METHODS

### Environment and data collection settings

Figure 1 shows the experimental environment. We used an elevator in our laboratory. The elevator has elevator car operating panels on both sides of the door. There were no mirrors or posters inside the elevator. It is 2.3 m high, 1.35 m wide, and 1.6 m deep. We installed a camera inside it and on the ceiling near the elevator's hallway to record video images during the experiment.

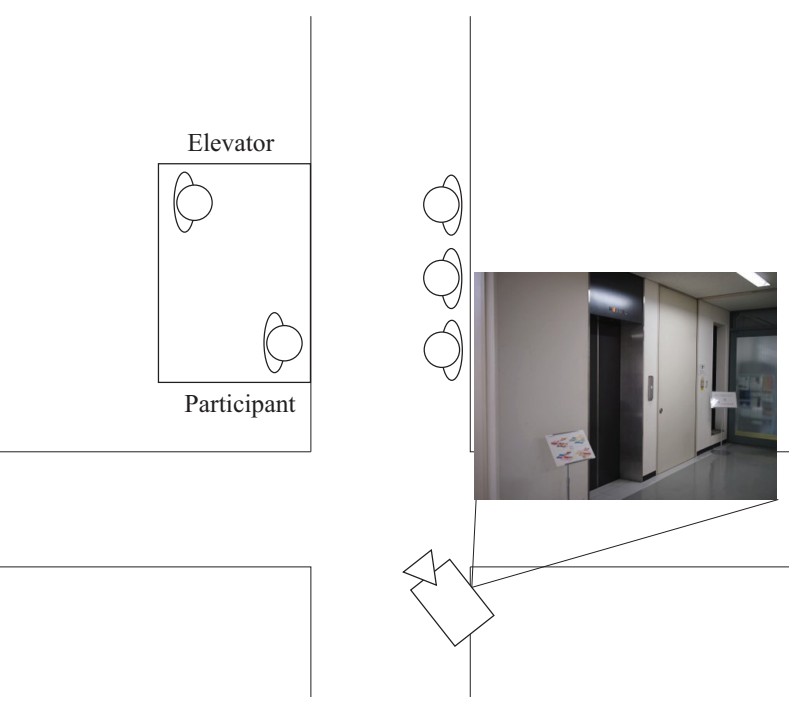

**Figure 1  Illustration of experiment environment.**  

## Participants

We recruited 30 participants (15 women and 15 men), all native Japanese speakers (mean age = 34.2 years, SD = 9.00), and conducted 48 trials with each participant (1,440 trials in total). The numbers of participants and trials were determined based on empirical heuristics and the constraints imposed by participants' limited time availability; *i.e.*, there was no *a priori* justification for the sample size. They were enrolled through a local commercial recruiting company and paid for their participation. All participants were physically healthy and had no mobility issues, resulting in similar riding times among individuals. For the data recording, they wore jackets with prominent numbers on them. The participants were strangers to each other and had no prior acquaintance before the experiment.

## Measurement

The participants measured their perceived waiting times (the *waiting time* was the interval beginning when the participant pressed the call-button in the lobby, and ending when the elevator arrived and its doors fully opened, and the *transit time* was the interval beginning when the participant stood the elevator, and ending when the elevator stopped and its doors fully opened at the destination floor) and their perceived stress during these two waiting situations. Because we continuously measured these perceived waiting times, we employed a prospective paradigm similar to past studies (*Lahera, 2022*). Such an approach is more accurate than a retrospective paradigm (*Brown, 1985*) and appropriate for multiple measurements.

We also measured the actual waiting times in seconds, the number of people waiting together, the number of times the elevator was missed during waiting, the number of people riding together, and the number of times the elevator stopped during transit, using recorded videos. The perceived stress was evaluated in an 11-point response format where 0 denotes no stress, and 10 is extreme stress. We employed an 11-point response format because a past study argued that using 11 scales closely approximated interval data (*Wu & Leung, 2017*).

## Procedure

All the procedures were approved by the Advanced Telecommunications Research Institute International Review Board Ethics Committee (525H). Written informed consent was obtained from all participants. Multiple participants simultaneously joined the experiment to reproduce various situations in elevators. We capped the number of participants (for any one situation) to five to prevent excessive congestion in the elevator even when all were riding at the same time. All participants first took the elevator on the third floor. They were provided with detailed instructions prior to the experiment; each participant received a printed sheet that illustrated the above timing rules, and a standardised oral briefing. Specifically, they were instructed to perform the following steps in sequence: (1) enter the elevator and select a predetermined floor number as instructed, (2) get off at the designated floor upon arrival, (3) refuse the elevator the next time it arrived (*i.e.*, deliberately skip it), and (4) wait for the subsequent elevator and then enter it, selecting another predetermined floor different from the previous one. In addition. they were not permitted to use timing devices, such as smartphones or watches, nor were they allowed to speak with other participants during the experiment. The primary reason for prohibiting smartphones was to prevent participants from accurately measuring waiting/transit times.

After the briefing, the participant completed one practice ride to a different floor. During this practice, the experimenter verified that the participant recorded the perceived waiting time and the perceived transit time as instructed. Only after successful practice did the experimental trials begin. Each participant completed 48 trials within a single session lasting less than 2 h. We asked the participants to write down their perceived waiting times and stress levels experienced while waiting for and riding the elevator on pre-printed article immediately after each ride. Through the experiment, each participant visited every destination floor the same number of times, but the order of descents varied across participants; the next destination floors were described on the pre-printed article. Within each group, we counterbalanced these orders to minimise potential order effects. We conducted the experiments in six groups, each consisting of five participants, over a period of 3 days, with two groups per day.

## RESULTS

We gathered 1,440 data sets (perceived waiting times, actual waiting times, and perceived stresses) from 30 participants. On average, 2.09 ± 0.97 participants (mean ± SD) shared the elevator. Occasionally, a few participants failed to record their waiting and transit times

during data collection, so we excluded these data. In addition, fewer than ten times external passengers entered the elevator, but we included these data in the analysis because they did not disrupt our experiment procedures. Finally, the amount of valid data for *waiting time* was 1,416 and 1,422 for *transit time*. With the gathered data, we analyzed the relationships among each type of data.

## Perceived and actual waiting/transit times

We analyzed the relationships between the perceived and actual waiting times for the *waiting* and *transit times*[1]. In this analysis, we focused on (1) whether the perceived waiting times were longer/shorter than actual waiting times, and (2) whether the repetition effects occurred in our experiment as reported in a past study (*Matthews, 2011*) that repeated trials tend to shorten perceived time durations, because the participants completed the elevator-riding task 48 times in this study. Therefore, we compared the average perceived/ actual *waiting* and *transit* times between the first and last halves of the trials.

Firstly, we conducted repeated two-factor repeated-measures ANOVA (*type* factor: *perceived* and *actual*, and *time* factor: *first* and *last*) for their average *waiting* time (Fig. 2), and the results showed that a significant main effect for the *type* factor ($F(1, 29) = 56.354$, $p < 0.001$, partial $\eta^2 = 0.660$). There were no significant main effects for the *time* factor ($F(1, 29) = 0.184$, $p = 0.671$, partial $\eta^2 = 0.006$) and the interaction effect ($F(1, 29) = 0.001$, $p = 0.990$, partial $\eta^2 < 0.001$).

We also conducted repeated two-factor repeated-measures ANOVA (*type* factor: *perceived* and *actual*, and *time* factor: *first* and *last*) for their average *transit* time (Fig. 3), and the results showed that a significant main effect for the *type* factor ($F(1, 29) = 51.425$, $p < 0.001$, partial $\eta^2 = 0.639$). There were no significant main effects for the *time* factor ($F(1, 29) = 1.718$, $p = 0.200$, partial $\eta^2 = 0.056$) and the interaction effect ($F(1, 29) = 0.556$, $p <= 0.462$, partial $\eta^2 = 0.019$).

In addition, we analyzed the correlation between the times in each situation. We found a statistically significant positive correlation for *waiting time* ($r = 0.510$, $p = 0.004$), but no significant correlation for *transit time* ($r = 0.332$, $p = 0.073$).

These results showed that the perceived waiting times were significantly shorter than the actual waiting times under our experimental conditions, and no evidence of repetition effects on estimated times within our experimental setting.

## Relationships between waiting/transit times and perceived stress

We analyzed the relationships between waiting times and perceived stress with SPSS and performed a linear mixed-effects analysis of the relationship between them[2].

First, we analyzed the relationships between waiting times and perceived stress (Figs. 4 and 5). The model included fixed effects for the number of people waiting together, the number of times the elevator was missed during waiting, and the perceived *waiting time* or actual *waiting time*, with repeated measures for trials and random effects for participants. The results indicated no significant effects for the number of people waiting together or the number of times the elevator was missed during waiting. Therefore, we used a model that only included fixed effects for the perceived *waiting time* or actual *waiting time*, with

[1] These analyses were modified in response to reviewer comments; in the original analyses, we separately investigated whether the perceived waiting times were longer/shorter than actual waiting times *via* the t-test with all datasets, and whether repetition effects exist *via* the t-test with the averaged dataset. The modified analysis showed similar patterns to the original ones, except for the correlation analysis, although we applied the Bonferroni correction.

[2] These analyses were also modified in response to reviewer comments; in the original analyses, we did not include participants as random effects. The modified analysis showed the same patterns as the original ones, although we applied the Bonferroni correction.
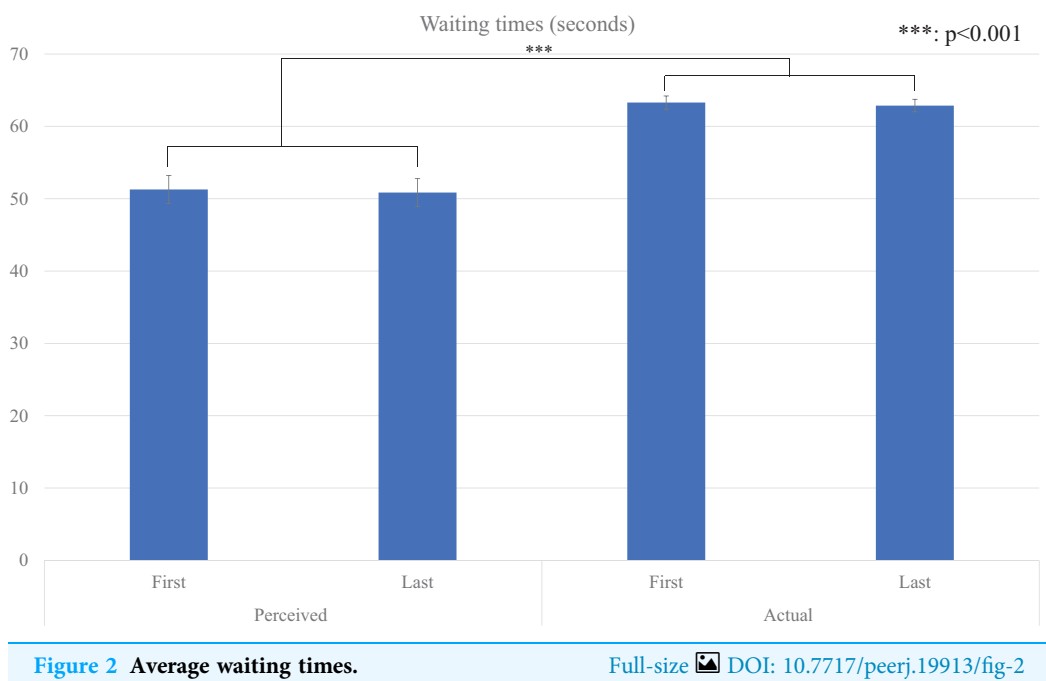

**Figure 2 Average waiting times.**

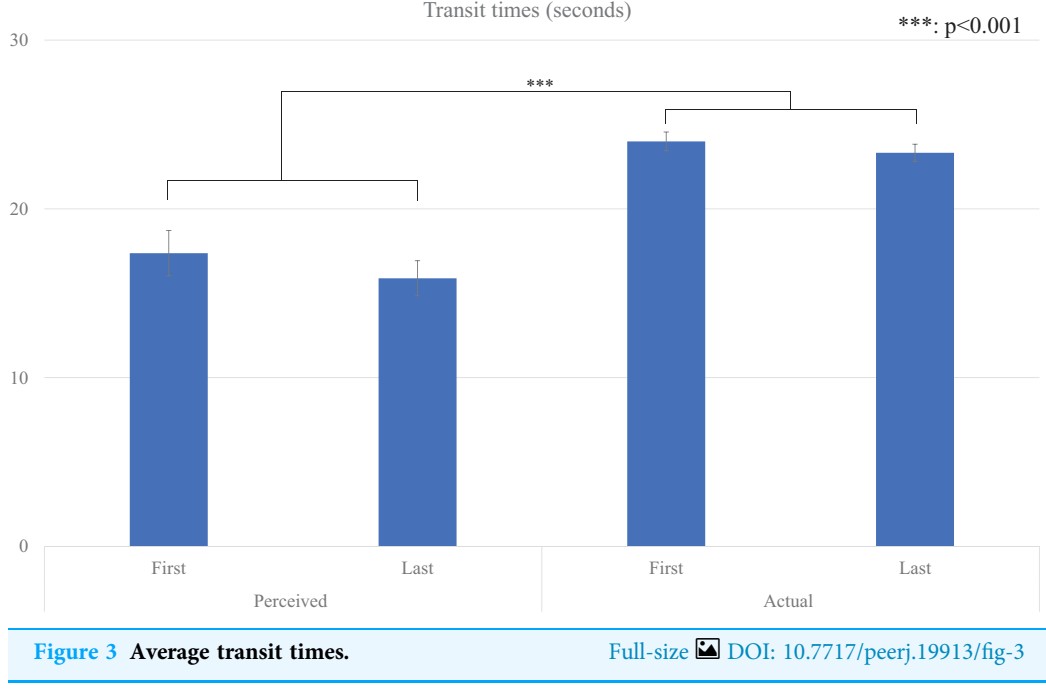

**Figure 3 Average transit times.**

repeated measures for trials and random effects for participants. The model was estimated using the Akaike Information Criterion (AIC). We compared the model fit indices between the perceived *waiting time* and the actual *waiting time*; the AIC showed that the model with the perceived *waiting time* showed a better fit (the perceived *waiting time*: AIC = 5,490.294, the actual *waiting time*: AIC = 5,767.842). A significant effect of the fixed

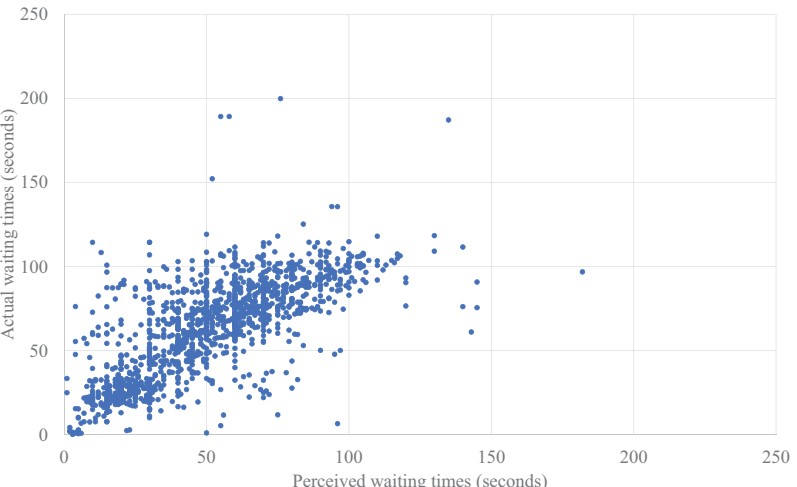

**Figure 4 Plotted data of perceived waiting time actual waiting time.**

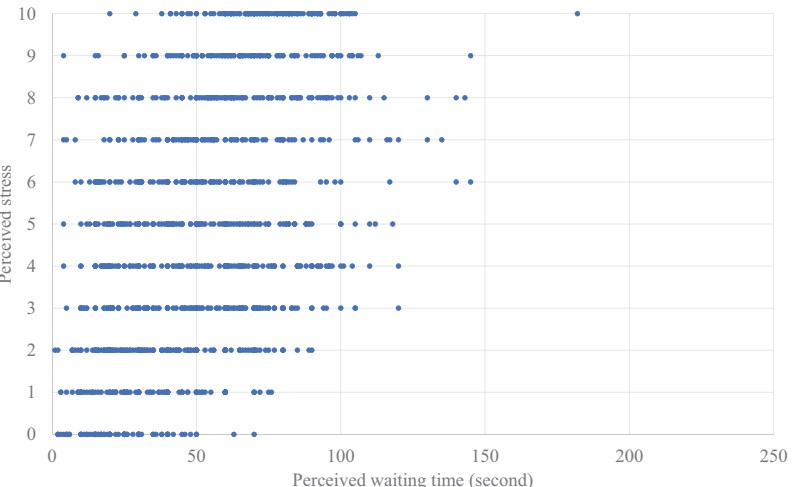

**Figure 5 Plotted data of perceived waiting time and perceived stress.**

effect for the perceived *waiting time* was also found ($F_{(1, 27.689)} = 164.372$, $p < 0.001$), although there was no significant correlation between the perceived *waiting time* and the perceived stress ($r = 0.180$, $p = 0.342$).

Overall, the estimated model of the perceived stress based on the perceived *waiting time* can be described as

$$Perceived\ Stress\ Of\ Waiting\ Time = 1.87 + 0.06 * Perceived\ Waiting\ Time. \qquad (1)$$

We also analyzed the relationships between the transit times and the perceived stress (Figs. 6 and 7) using the same analysis procedures with a linear mixed-effects analysis. The model included fixed effects for the number of people riding together, the number of times

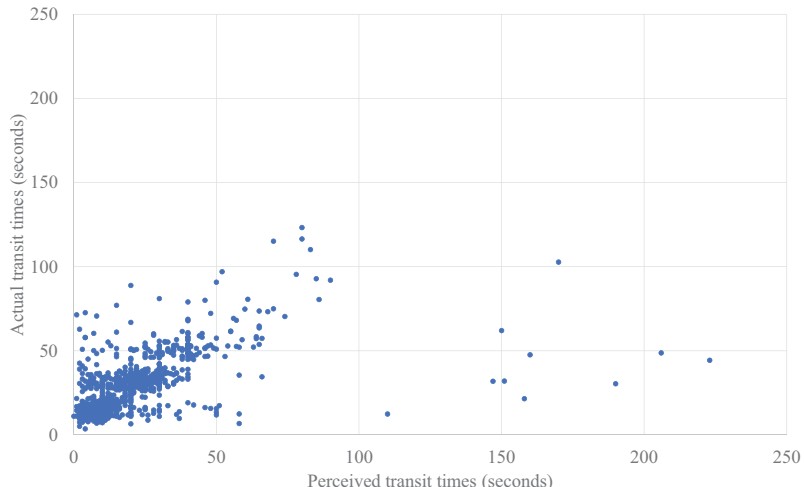

**Figure 6 Plotted data of perceived transit time actual transit time.**

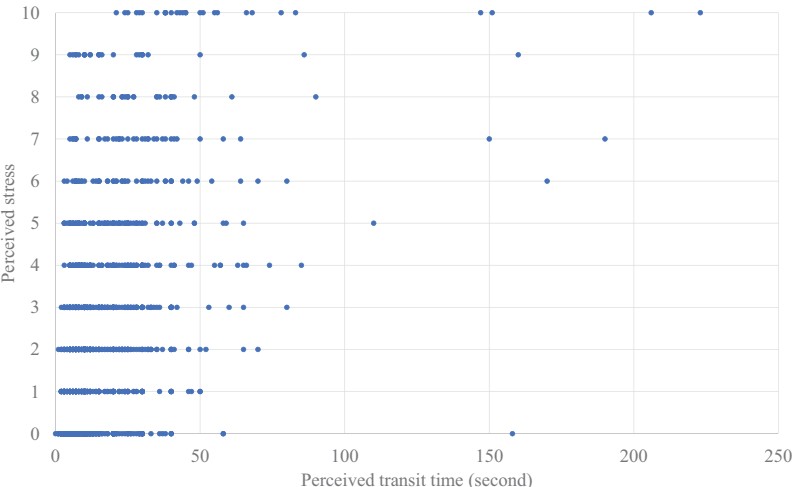

**Figure 7 Plotted data of perceived transit time and perceived stress.**

the elevator stopped during transit, and the perceived *transit time* or actual *transit time*, with repeated measures for trials and random effects for participants. The results showed that the number of times the elevator was missed during waiting was not significant for either perceived or actual transit times. After excluding this non-significant variable, we compared the model fit indices for perceived *vs*. actual *transit times* by using the AIC. The AIC showed that the model with the perceived *transit time* showed a better fit (the perceived *transit time*: AIC = 5,252.486, the actual *transit time*: AIC = 5,610.283). Significant effects were found for the perceived *transit time* ($F$ (1, 23.947) = 62.579, $p < 0.001$) and the number of people riding together ($F$ (1, 6,043.062) = 13.039, $p < 0.001$), although there was no significant correlation between the perceived *transit time* and the

perceived stress ($r = -0.080$, $p = 0.673$). Overall, the estimated model of the perceived stress based on the perceived *transit time* and the number of people riding together can be described as

$$Perceived\,Stress\,Of\,TransitTime = 0.45 + 0.10 * Perceived\,TransitTime \\ + 0.22 * Num\,Of\,Passenger \qquad (2)$$

## DISCUSSION

### Perceived waiting times were shorter than actual waiting times

Unlike past studies (*Fan, Guthrie & Levinson, 2016*), the perceived waiting times were significantly shorter than the actual waiting times in each waiting situation (elevator's arrival and riding it) under our specific experimental conditions. A possible reason might be that we asked the participants to record their perceived waiting times and perceived stress only in each waiting situation, *i.e.*, situations with relatively small cognitive loads. Perhaps such simple tasks enabled them to focus on temporal measurements. Moreover, we employed a prospective paradigm to measure perceived waiting times; a past study reported that participants felt shorter durations under small cognitive loads, and the prospective paradigm provided shorter time lengths than the retrospective paradigm (*Brown, 1985*).

One possible explanation for the shortened perceived time is the participants' attention directed toward the passage of time. Previous studies have indicated that reduced attention to the passage of time can lead to shorter or inaccurate estimations (*Brown, 1985*; *Sawyer, Meyers & Huser, 1994*). However, participants were not permitted to use timing devices, such as smartphones or watches, nor were they allowed to engage in conversations with other participants during the experiment. Indeed, our video analysis confirmed the absence of conversational interactions. Thus, we assume that participants' attention was adequately focused on the experimental task itself.

### Perceived stress estimation

Our linear mixed-effects analysis showed the effectiveness of using perceived waiting times to estimate perceived stresses compared to actual waiting times. This finding is consistent with previous studies that reported relationships between perceiving *waiting time* and stress (*Abdul Hamid, 2021*; *Ayodeji, Rjoub & Özgit, 2023*; *Bielen & Demoulin, 2007*; *Pruyn & Smidts, 1998*). Another study reported that passengers preferred a longer *transit time* than a longer *waiting time* (*Bird et al., 2016*); our equations also support this phenomenon.

Equation (1) for perceived stress while waiting indicated that the stress value approaches the middle value (five) after waiting for 1 min. This suggests that a majority of passengers feel stress when the *waiting time* exceeds 1 min. In a web survey of 400 people by a Japanese watch company (*CITIZEN, 2023*), 66.5% of the respondents felt stressed when they had to wait for more than a minute for an elevator, a result that resembles those shown by our model.

In addition, Eq. (2), which describes perceived stress during riding, indicates that stress levels were influenced not only by the transit time but also by the number of passengers. It is plausible that the limited space inside the elevator, compared to the lobby, increased stress due to the proximity of other passengers. Moreover, the coefficient indicating the impact of increased transit time on stress is greater than that for waiting time in Eq. (1), possibly due to the confined, enclosed nature of the elevator environment, which may cause an invasion of personal space (*Hall, 1966*) by close physical proximity to other passengers.

As an additional analysis, we examine whether repetition effects occurred in our experiment in the context of perceived stress, similar to the perceived/actual *waiting* and *transit* times analysis. For this purpose, we also compared the average perceived stress levels between the first and last halves of the trials using paired t-tests. The results revealed no significant differences in the perceived stress during waiting ($t(29) = 0.364$, $p = 0.719$, $d = 0.066$) and the perceived stress during riding ($t(29) = 0.670$, $p = 0.508$, $d = 0.122$). Thus, we found no evidence of repetition effects on perceived stress levels.

### Reducing stress in elevator riding

From the perspective of reducing the stress of those taking elevators, an engineering approach, which improves the movement efficiency of elevators, *i.e.*, shortening objective waiting times, can indeed be effective; however, a cognitive approach that reduces the perceived waiting times may be even more beneficial. For example, the installation of mirrors or televisions in elevator lobbies, as reported in past studies (*Pruyn & Smidts, 1998*; *Sasser et al., 1978*), are one cognitive strategy to reduce perceived waiting times. Furthermore, since waiting-time satisfaction has been shown to mitigate stress (*Bielen & Demoulin, 2007*; *Nah, 2004*), providing information to passengers waiting for an elevator by devices represents another useful method to enhance satisfaction during waiting periods. From this perspective, both equations are useful for managing the appropriate contents for passengers. For example, an elevator system can identify overly frustrated passengers by tracking them, measuring their waiting times, and changing their information contents depending on such passenger characteristics as age, gender, and demeanor. Such an approach, which resembles recent advertisement systems to attract them (*Alhalabi et al., 2021*; *Sewmina, Navarathna & Senanayake, 2023*), might decrease their perceived waiting times.

### Limitations

This study has several limitations. First, the experiment has a simple setting in a building with just four stories. Hence our experiment settings cannot cover situations that require longer waiting times, such as in high-rise buildings. The elevator's size is also fixed, and the experiment was only conducted in Japan. Additional investigations on different kinds of elevators and cultures are essential to generalize the knowledge from this experiment.

We did not confirm the participants' abilities in time perception in advance; therefore, the experiment results might have been influenced by their characteristics. In addition, we did not explicitly prohibit counting time on their heads. Therefore, it is possible that silent

counting influenced their time estimates. However, at the same time, participants had to note their perceived time and stress on article, and monitor the floor on which to disembark, which may have implicitly reduced the likelihood of sustained counting. Finally, no *a priori* power analysis was conducted to justify the sample size; we acknowledge this explicitly for transparency (*Lakens, 2022*; *Sasaki & Yamada, 2023*). We likewise refrain from including a *post-hoc* power analysis because of its well-documented limitations (*Hoenig & Heisey, 2001*). Future studies that incorporate pre-registered power calculation (*e.g.*, using G\*power (*Faul et al., 2007*)) and larger samples will allow more rigorous statistical interpretation.

## CONCLUSIONS

This study investigated the gaps between perceived and actual waiting times as well as the relationships between both times and the perceived stress while passengers used elevators. We conducted a data collection experiment where participants reported their perceived waiting times and perceived levels of stresses in elevator-use situations. The data analysis results showed that perceived waiting times are significantly correlated with actual waiting times and are significantly shorter than the actual waiting times for both the *waiting time* and the *transit time* under our experimental conditions. The perceived stress was also better modeled using the perceived waiting times than the actual waiting times. In addition, the perceived stress while waiting for an elevator exceeded the perceived stress while taking one. The knowledge from our study will contribute to understanding the time perceptions of elevator passengers and designing human-elevator interactions to reduce their perceived stresses.

## ACKNOWLEDGEMENTS

We thank Sayuri Yamauchi for their help during our experiments. ChatGPT (OpenAI, ChatGPT 4.5) was used for the initial English language editing and proofreading of the manuscript. The final version of the manuscript was reviewed and further edited by a professional editing service, whom we thank for their assistance.

### Funding

This work was funded by the Mitsubishi Electric Corporation and JST CREST (Core Research for Evolutional Science and Technology) Grant Number JPMJCR18A1. The funders had no role in study design, data collection and analysis, decision to publish, or preparation of the manuscript.

### Grant Disclosures

The following grant information was disclosed by the authors:
Mitsubishi Electric Corporation and JST CREST (Core Research for Evolutional Science and Technology): JPMJCR18A1.

## Competing Interests

Masayuki Kakio is employed by Mitsubishi Electric Corporation. Other authors declare that they have no competing interests.

## Author Contributions

- Masahiro Shiomi conceived and designed the experiments, performed the experiments, analyzed the data, prepared figures and/or tables, authored or reviewed drafts of the article, and approved the final draft.
- Masayuki Kakio conceived and designed the experiments, performed the experiments, authored or reviewed drafts of the article, and approved the final draft.
- Takahiro Miyashita conceived and designed the experiments, authored or reviewed drafts of the article, and approved the final draft.

## Human Ethics

The following information was supplied relating to ethical approvals (*i.e.*, approving body and any reference numbers):

This research was approved by The Advanced Telecommunications Research Institute International Review Board Ethics Committee (525H).

## Data Availability

Raw data is available in the Supplemental Files.

## Supplemental Information

Supplemental information for this article can be found online at http://dx.doi.org/10.7717/peerj.19913#supplemental-information.

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
