# Peer review of "How long is too long? Examining waiting times and stress in human-elevator interaction"

_PeerJ, doi:10.7717/peerj.19913_

## Round 0.1 · original submission · Major Revisions

· Academic Editor

Major Revisions

Thank you for submitting your work to PeerJ. Of the two reviewers, one specializes in basic research on time perception, while the other focuses on applied research related to time.

Before addressing the details of their comments, I must inform you that your manuscript will require major revisions before it can be considered for publication. Both reviewers expressed concerns about the clarity of the experimental description, noting that insufficient information was provided. I share this concern. Additionally, it would be helpful to explain how the sample size was determined. Reviewer 2 also commented on the strength of the claims made in the manuscript.

In line with PeerJ’s publication criteria, these issues must be thoroughly addressed in order to make the manuscript suitable for publication.

Reviewer 1 ·

Basic reporting

This study experimentally examined the relationship between perceived waiting time and stress in an elevator usage scenario. The experiment is straightforward, and the conclusions are clear. However, the study requires revision, particularly in providing details on the experimental conditions, as it lacks essential information needed to fully understand the experiment.

Experimental design

1. Describe in detail the instructions given to the participant on line 184 of the text.

2. Related to the first comment, please describe whether or not participants were allowed to handle their smartphones or talk with other participants during their participation in the experiment.

3. Describe whether the 30 participants were gathered together at one time or divided into multiple groups for the experiment. Also, please describe whether the 48 trials conducted by the participants were done in a single day or divided over several days.

4. Clarify whether or not the participant had permission to count time while participating in the experiment.

Validity of the findings

5. A correlation figure between stress and estimated waiting time seems necessary.

6. A correlation analysis between the number of people who rode the elevator together (or waited together) and stress and estimated time would also be needed.

7. A possible reason for the shortened estimated time could be the factor of trial repetition. Previous studies have reported the phenomenon that repetition of trials shortens the perceived time (e.g., Matthews, 2011), and it is possible that the 48 repetitions had an effect in this study as well.

Matthews, W. J. (2011) Stimulus repetition and the perception of time: the effects of prior exposure on temporal discrimination, judgment, and production. PLoS One, 6(5), e19815.

8. Another possible reason for the shortened estimated time could be the influence of attention directed to the passage of time. Previous studies have shown that a decrease in attention directed to the passage of time results in shorter or inaccurate estimated time (e.g., Brown, 1985; Sawyer, Meyers, & Huser, 1994). For example, if participants were conversing with each other, it is possible that time was rated shorter because conversation reduced the attention directed to the passage of time.

Brown, S. W. (1985) Time perception and attention: The effects of prospective versus retrospective paradigms and task demands on perceived duration. Perception & Psychophysics, 38, 115–124.

Sawyer, T. F., Meyers, P. J., & Huser, S. J. (1994) Contrasting task demands alter the perceived duration of brief time intervals. Perception & Psychophysics, 56, 649–657.

Additional comments

9. In line 191 of the text, write the reason why the data were not validated.

10. Describe whether or not there were mirrors or posters in the elevator used in this experiment.

11. It seems to me that the estimated wait times and stresses are different in the first half of the 48 trials and in the second half. I would suggest analyzing this.

Reviewer 2 ·

Basic reporting

There is a long history of research on time perception, and various findings have been reported, but there are very few citations of relevant research in this study. For example, the following citations reporting on the discrepancy between actual time and perceived time are missing.

- L. Bufadri. 1971. Factors affecting the filled-duration illusion in the auditory, tactile, and visual modalities. Perception & Psychophysics 10, 4B (1971), 292–294.
- S-S. Mo. 1975. Temporal reproduction of duration as a function of numerosity. Bulletin of the Psychonomic Society 5 (1975), 165–167.
- D. Zakay and R. A. Block. 1996. The Role of Attention in Time Estimation Processes. Elsevier Science, Amsterdam, 143–164.

Experimental design

The actual method for measuring the actual waiting time and perceived waiting time when using the elevator is unclear. Is the waiting time the duration from when a participant exits the elevator to when they board the next elevator, or is it the duration from when they join the queue for the elevator to when they board the elevator? Also, is the transit time from when they get on the elevator to when they reach their destination included in these waiting times? And how did the participants receive the instructions from the experimenter on how to measure their perceived waiting time? These details are not described.

As the authors also mention in the section "limitations," although this experiment used a real elevator to observe the participants' perceived waiting time and their stress, it seems to be very different from the actual environment in our daily lives. For example, most of the stress in boarding an elevator should be derived from the following situations: not being able to get on even if you are in line, or the elevator stopping many times before reaching the destination floor. However, it appears that these situations have not been simulated. Therefore, it is unclear what factors affected the participants' stress in this experiment (longer waiting times lead to more stress is too obvious).

Validity of the findings

It is understandable that the relationship between the actual waiting time and the perceived waiting time is understood by using the correlation coefficient, but does the intention to compare the mean values of these using the t-test match the purpose of this paper? Do they want to argue that the predicted time will always be shorter than the actual time?

Additional comments

Although the authors' study focusing on the relationship between the perceived waiting time, the actual waiting time when entering and existing an elevator, and their stress is quite interesting and unique, the experiment in this paper is still in the preliminary stage, and it can be said that it is premature to be published in a journal article.
I then recommend that the authors redesign the experiments to imitate a more realistic situation of the elevator used.

---

## Round 0.2 · Major Revisions

· Academic Editor

Major Revisions

Thank you for submitting the revised version of your manuscript to PeerJ. We appreciate your efforts in addressing the initial concerns. The revisions have improved the clarity of the experimental procedures and analytical methods, which has significantly enhanced the readability and interpretability of the work.

Please note that one of the original reviewers declined to assess the revised manuscript. Therefore, we invited a new reviewer with expertise in time perception to ensure a thorough evaluation.

With the improved clarity in the methodology and analyses, it is now easier to understand the structure and rationale of the study. At the same time, this has made it more apparent that some important issues remain unresolved. These issues must be carefully addressed to meet the standards for publication.

Based on the current assessment, the decision remains Major Revision. However, we believe that with adequate revisions, the manuscript has the potential to become suitable for publication.

Both Reviewers 1 and 2 have provided detailed and critical feedback, particularly regarding your analytical approach. We strongly encourage you to respond to these points thoroughly in your next revision.

One additional note: If you decide to revise your analysis or statistical tests in accordance with the reviewers’ suggestions, please be mindful that such changes represent a deviation from your original analytical plan. While such changes may be scientifically meaningful, they also raise concerns about inflated Type I error rates. To address this, we recommend ensuring transparency—either by clearly stating in a footnote that “This analysis was modified in response to reviewer comments; the original analysis was [brief description],” or by reporting the revised analysis as an additional exploratory analysis rather than a replacement of the original. In either case, readers should be able to distinguish between confirmatory and exploratory components of the work.

We look forward to receiving your revised manuscript.

Reviewer 1 ·

Basic reporting

The author’s revisions enabled me to interpret the methods and results of the experiment. However, I believe that the following points still need to be addressed.

Experimental design

On counting in the head during the experiment:
In studies of time perception, participants are sometimes instructed, “Please do not count in your head during the experiment”. Was such an instruction given in this study?

How to respond to waiting times:
Please clearly describe how participants responded to the waiting time (probably by writing it down on paper). Additionally, provide a clear explanation of the instructions that were given to participants for this procedure.

Validity of the findings

A priori power analysis:
Please clarify the basis for the effect size and the planned statistical test used in the a priori power analysis described in lines 166–167.

Handling of data within and between participants:
If my understanding is correct, the 1,440 data sets used in this study include a mixture of data at different hierarchical levels, such as within-individual and between-individual data. Therefore, Therefore, I believe that hierarchies should also be incorporated into the statistical test models used in this study. If a hierarchical model is not adopted, the rationale for this decision should be clearly explained.

Reviewer 3 ·

Basic reporting

The current study addressed the relationship between perceived waiting time and stress, which is an interesting and relevant topic in real life. While I believe the results could contribute to existing literature to a certain extent, I have some concerns, as I described below.

Experimental design

There are many unclear points in the method. How were participants' perceived time and stress recorded (written on paper)? How were the designated floors determined, and how were participants informed about them? Did all five participants get on the elevator at the same time? Was anyone other than the participants riding in the elevator? Were there differences in the actual time spent riding and waiting among participants and groups? Please provide more detailed information on the specific procedures so that replication studies can be conducted.

In the current analysis, it appears that correlation coefficients were calculated using data collected across multiple trials per participant. This raises concerns about violating the assumption of independence between observations. Traditional correlation analyses assume that all data points are independent, and violating this assumption can result in inflated Type I error rates. To address this issue, I strongly recommend using mixed-effects modeling approaches, such as Linear Mixed Models (LMMs), with participants as random effects. These models are well-suited for nested data structures and can appropriately account for within-subject dependencies by including random effects for participants.
Moreover, although LMMs were used in analyses examining the relationship with stress, it would be better to include participants as random effects in this case as well.

T-tests were used to compare the actual and perceived waiting times, but judging from the degrees of freedom, it appears that the data for each participant was being used as it was in the analysis. If so, it is necessary to calculate the average value for each participant and use them for analysis.

Validity of the findings

I’m concerned that it is difficult to draw valid conclusions about the difference between actual and perceived waiting times because it was not confirmed in advance that the participants were able to correctly evaluate the actual time. In the field of time perception, it is common practice to compare the perceived time between conditions (experimental condition vs. control condition). If participants had a tendency to evaluate the time as shorter than it actually was, then the results simply reflect that tendency. This point should be noted as a limitation.

Additional comments

The figure numbers do not seem to match. Figure 2 in the text appears to refer to Figure 6.

---

## Round 0.3 · Minor Revisions

· Academic Editor

Minor Revisions

Thank you for resubmitting your manuscript. I shared the revised version with the reviewers from the previous round. Both reviewers were generally satisfied with your revisions. However, Reviewer 2 raised a remaining concern. I agree with the concern itself, but I do not think the suggested solution is ideal. That said, this issue can be resolved with a minor revision, and therefore my decision is “Accept subject to revision.” I explain the details below.

In your revised manuscript, you re-conducted a power analysis and described it as an “a priori” power analysis in the Participants section. Reviewer 2 pointed out that this is inappropriate, and also noted that G*Power calculates sample size based on the number of participants, not the number of trials. I agree with both of these points, particularly the former. Conducting a power analysis after data collection and presenting it as a priori is problematic, and arguably falls into what has been described as SPARKing (Sasaki & Yamada, 2023).

To address this issue, Reviewer 2 suggested reporting a post hoc power analysis. However, post hoc power analyses have well-known limitations and are generally discouraged in this context (e.g., Hoenig & Heisey, 2001; Lenth, 2007). Therefore, I would recommend avoiding the use of post hoc power altogether.

Given the circumstances, I believe the most transparent and responsible approach would be to state that there is no a priori justification for the sample size. This should be acknowledged explicitly, and the tone of any related claims should be correspondingly modest. For further guidance, see Lakens (2022), who advocates for such transparency when power justification is not available.

Please revise the relevant portion of the manuscript accordingly. Once this minor revision is completed, we will proceed to formally accept the paper.

Reviewer 1 ·

Basic reporting

The author has responded thoughtfully to the comments and suggestions made in the previous review round. The revisions have improved the clarity and overall quality of the manuscript. I believe that the revised manuscript is now suitable for publication. There are no further significant concerns.

Experimental design

no comment

Validity of the findings

no comment

Reviewer 3 ·

Basic reporting

The authors responded appropriately to my comments and the revision has substantially improved the manuscript. One thing I'm still concerning about, however, is the description in the Participants section.
In the revised manuscript, the authors have reworked the power analysis, yet it is still described as "a priori." In addition, sample size in G*power refers to the number of participants, not the number of trials. I recommend performing a post hoc power analysis using the obtained data and the actual number of participants and reporting the power of the analysis of interest.

Experimental design

no comment

Validity of the findings

no comment

---

## Round 0.4 · accepted · Accept

· Academic Editor

Accept

Thank you for resubmitting your manuscript. As the previous round involved only minor issues, I did not send the revised manuscript back to the reviewers. I have reviewed the revision myself and am largely satisfied with the changes. Therefore, my decision is “Accept.”

However, I would like to request one final minor revision. In the revised manuscript, you wrote the following regarding power analysis:
“Although we re‑ran a power analysis for internal reference, presenting those results could be considered SPARKing (Sasaki & Yamada, 2023), so we have chosen not to report them.”

This reflects a misunderstanding of the SPARKing issue. It is not problematic to conduct a power analysis after data collection and report it transparently for internal reference. What constitutes SPARKing (Sasaki & Yamada, 2023) is presenting such an analysis as if it were a priori, when in fact it was conducted post hoc.

I encourage you to reread my previous comment and our paper (Sasaki & Yamada, 2023) for clarification. Accordingly, please either revise or remove the above sentence. In my view, simply omitting the sentence would be perfectly acceptable.

Once this minor edit is made, your manuscript will be ready for publication.